# Theoretical and Experimental Investigation on the Flexural Behaviour of Prestressed NC-UHPC Composite Beams

**DOI:** 10.3390/ma16020879

**Published:** 2023-01-16

**Authors:** Pengzhen Lin, Weiyi Yan, Hongwei Zhao, Junjun Ma

**Affiliations:** 1Department of Civil Engineering, Lanzhou Jiaotong University, Lanzhou 730070, China; 2China Railway Design Corporation, Tianjin 300450, China

**Keywords:** ultra-high-performance concrete, NC-UHPC composite beams, bending resistance test, calculation of bending capacity

## Abstract

To investigate the normal section strength and cracking bending moment of normal concrete–ultra-high-performance concrete (NC-UHPC) composite beams, calculation formulas were established considering the tensile strength of UHPC based on the current railway bridge design code. Using the railway T-beam as a template, prestressed NC-UHPC composite beams with different NC layer heights were built. A static bending test was performed, the pressure of the steel strand and the deflection and strain of the beam were measured, and the evolution of cracks in each beam was observed. The calculation formulas of the normal section strength and cracking bending moment of NC-UHPC composite beam were verified by the test. The results showed that the type of strain was similar to load-deflection curves with increasing load; the bending failure process of the NC-UHPC composite beam showed four obvious stages: elasticity, uniform cracking, crack development, and yield. Cracks in the beam started to appear at stage II, developed rapidly at stage III, and stopped emerging at stage IV. The calculation formulas for the normal section strength and the cracking bending moment of the NC-UHPC composite beam were in good agreement with the test values. Normal concrete with a compressive strength of 80 MPa can replace UHPC for the design of NC-UHPC composite beams.

## 1. Introduction

Ultra-high-performance concrete (UHPC), one of the most popular new cement-based composites of the 21st century, has been welcomed by the bridge industry for its excellent mechanical properties and durability, and has wide-ranging application prospects in the bridge field [1]. After decades of theoretical research, model tests and engineering trials, it is clear that UHPC members significantly reduce structural weight and can greatly improve the durability and span of bridges [2].

Flexural behavior is an important characteristic of concrete bridges. Research has shown that prestressed UHPC girders demonstrate excellent ductility, cracking resistance, and flexural deformation capacity [3] due to the use of fibers, which significantly improve the strength, toughness, and durability of UHPC. The flowability of UHPC mixtures is significantly reduced due to the use of steel fiber [4], while the compressive strength and flexural properties are considerably improved with increase in steel fiber content [5,6]. However, when the content exceeds 2%, the effect on strength and toughness is limited [7]. Due to the addition of steel fibers, the entire stress failure process of reinforced UHPC beams under flexural load is different from that of normal concrete (NC) and can be roughly divided into three stages [8,9]. The first is the elastic stage in which the bending moment in the pure bending section is small, the stress is basically the same as that of a uniform elastic body, and the load-deflection curve is a straight line. After an initial crack in UHPC, a crack development stage occurs in which the original tensile force carried by the UHPC is transferred to the steel fiber and the tensile longitudinal reinforcement. The slope of the deflection curve decreases gradually. In the later stage of damage, the longitudinal tensile reinforcement yields, and the compressive zone gradually starts to enter the stage of elasticity. The cracks in the purely bending span develop rapidly along the height of the beam and increase in width, the section reaches its ultimate load, the slope of the load-deflection curve gradually tends to zero, and an obvious main crack appears. The ratio of the elastic stage to the entire failure process of the UHPC beam is greatly increased and the ratio of the cracking load to the yield load is also increased [10].

The flexural behavior of prestressed UHPC girders is significantly superior to that of NC girders for a similar cross-sectional geometry [11]. However, the cost of raw materials, such as quartz sands and steel fibers, in a UHPC structure is higher and construction costs are greatly increased by using UHPC to build the whole structure [12]. Furthermore, existing research indicates that, when bending failure occurs, the stress in the compressed zone is much less than the compressive strength of the UHPC, which cannot be fully used [13]. There are two options for dealing with this issue: to insert more steel reinforcements or prestressed tendons in the tension zone to increase the bearing capacity of the beam, or to replace UHPC with NC in the compression zone. For the first option, the beam must be designed to have sufficient capacity, adding reinforcement will cause waste, and too much reinforcement in the tension zone will cause the beam to suffer over-reinforcement damage, which is harmful to the beam. The second option can reduce costs while ensuring the capacity of the beam. Therefore, based on the whole cast UHPC beam, UHPC in the compression zone can be replaced by NC, which has relatively low compressive strength.

In recent years, a considerable amount of experimental research has been undertaken on the flexural behavior of NC-UHPC composite beams, including that reported in [14,15,16,17,18]. In [14], the bond strength between UHPC and NC was found to be very high. When a flexural load was applied, an initial crack was observed to occur diagonally in the mid-depth of the shear span of the NC layer. Additional diagonal cracks were formed within the shear span as the applied load increased. Failure occurred when the concrete was crushed at the NC layer [14]. The initial and yield stiffnesses, as well as the peak and ultimate loads, were found to be enhanced by increasing the thickness of the UHPC layer, enabling expression of its high strength and ductility characteristics [15]. Compared with an NC beam, the numbers of cracks in a UHPC-NC composite beam was found to be significantly reduced [16]. NC-UHPC composite beams were shown to exhibit three failure patterns, including typical flexural failure, a hybrid of debonding and NC overlay flexural failure, and a hybrid of debonding and NC overlay shear failure [17]. The interfacial zone of a UHPC-NC composite beam does not affect the cooperative performance of the concrete beam and the UHPC layer before the failure stage. Therefore, the concrete beam and the UHPC layer can be understood to operate closely together. However, at the ultimate stage, due to a large number of fine cracks in the interfacial zone, the concrete will be softened, and the stiffness of the interfacial zone will be reduced, which represents one of the main failure modes of the composite structure [18].

In addition to experimental investigations, numerical and theoretical analyses are also effective approaches to the study of the flexural behavior of NC-UHPC composite beams. With theoretical research, and the development of calculation formulas, the reliability of experimental results obtained can be verified and, at the same time, a reference for engineering design can be provided [19]. Using numerical simulations, such as finite element models, reliable predictions can be provided in the absence of experimental data [20,21]. In numerical simulations, the tensile strength of concrete can be considered by defining material parameters. If this is ignored in theoretical calculations, it may result in significant deviations and cause the structure to be damaged by over-reinforcement when predicting the bending response of UHPC structures [22]. Therefore, the contribution of the tensile strength to the normal section strength and the cracking moment of UHPC structures is addressed in this study. Using six beams with three different heights of NC, static bending tests were carried out to explore the bending strength of NC-UHPC composite beams. In addition, in applied engineering, bridge structures need to have a safety margin, which is usually defined by the safety factor (the ratio of ultimate value to design value) [23]. In this paper, the test results are compared with theoretical calculation results and the safety margin for NC-UHPC composite beams is discussed.

## 2. Theory and Calculation

### 2.1. Calculation Method of UHPC Tensile Strength

Figure 1 depicts a proposed model for the stress-strain relationship of UHPC [24]. For the compressive side, when the stress reaches ultimate compressive stress *σ*_bcu_, it stops increasing, but the strain keeps growing until ultimate strain *ε*_u_ occurs. For the tensile side, when the first crack appears at elastic strain *ε*_e_, the slope of the curve begins to decrease; the stress reaches its peak at ultimate tensile stress *σ*_btu_ when the tensile strain corresponds to a crack width of 0.3 mm (*ε*_u0.3_). Then, the stress begins to decrease, and the strain keeps growing until *ε*_lim_.

UHPC has high tensile strength, ultra-high toughness, and huge ultimate tensile strain [25,26,27]. As can be seen from Figure 1, due to the addition of steel fibers, UHPC retains some of its tensile ductility and high residual strength after tensile cracking [28]. The fiber and matrix of UHPC material are regarded as two distinct materials based on the theory of composite mechanics. The tensile strength of UHPC is thought to derived from the UHPC matrix and fiber. Its value can be calculated by the sub-section function provided in Equation (1) based on the condition of the UHPC structure.
(1)$$f_{t,p}=\left\{\begin{array}{cc} f_{m}\left(1-V_{f}\right)+\alpha \lambda \tau_{a}\left(l_{f}/d_{f}\right)V & \mathrm{before}\;\mathrm{cracking}\\ \alpha \lambda \tau_{a}\left(l_{f}/d_{f}\right)V_{f} & \mathrm{after}\;\mathrm{cracking} \end{array}\right.$$
where *f*_t,p_ is the ultimate tensile strength of UHPC; *f*_m_ is the ultimate tensile strength of the UHPC matrix; *V*_f_ is the volume content of steel fiber; *α* is the effective direction coefficient of steel fiber, taken as 0.35 [29]; *λ* is the calculation parameter related to the fiber shear length, taken as 2 [30]; *l*_f_/*d*_f_ is the slenderness ratio of the steel fiber; and *τ*_a_ is the average bonding shear stress between the steel fiber and the UHPC matrix. The relationship between *τ*_a_ and the axial ultimate compressive strength *f*_c,p_ of UHPC can be represented as Equation (2) [31]:(2)$$\tau_{a}=0.60\sqrt{f_{c,p}}$$

### 2.2. Normal Section Strength of Prestressed NC-UHPC Composite Bridge

Many studies have been conducted on the flexural performance of cast-in-place UHPC beams [32,33,34,35,36]. The results of these studies indicate that, when UHPC beams are under flexural load, the compressive performance of UHPC in the compression zone cannot be fully used. Therefore, costly UHPC is substituted by normal concrete with sufficient compressive strength to meet the various requirements of the project for structural safety and economy.

The following assumptions are made for an NC-UHPC composite beam: (i) It can be considered that UHPC and NC have a reliable connection when calculating the normal section strength. Studies have shown that the bonding between UHPC and NC can be established when the interface is rough enough and that relative sliding can be ignored [37]. The strain on the section remains flat [38] and, if the interface between the UHPC and the NC is connected reliably, the relative slip can be ignored [39]; (ii) When the beam reaches its maximum strength, the longitudinal reinforcement deforms within the elastic range, the bond between UHPC and the steel strands is regarded as reliable, and the slip is ignored; (iii) The compressive strength of UHPC is greater than that of NC, so UHPC in the compression zone has not yet reached the ultimate compressive strain and can effectively help in the compression of NC before the NC is crushed at the top of the damage section. As a result, normal concrete and UHPC in the compression zone are all treated as normal concrete; the normal stresses are equal to rectangles for NC in the compression zone and UHPC in the tension zone [40].

Unlike cast-in-place concrete beams, which have section types based on the relative position of the neutral axis and the web, NC-UHPC composite beam sections can be classified into six types based on the relationship between the height of the compression zone of the section *x*_0_, the thickness of the upper flange *h*′_f_, and the height of the NC layer *h*_1_. Figure 2 depicts the classification of the six sections.

Taking the III-I section as an example, in which *h*_1_ > *h*′_f_ > *x*_0_. When the NC-UHPC composite beam achieves the ultimate strength, the distribution of strain and equivalent force are as shown in Figure 3.

The normal section strength of the I-beam depicted in Figure 3 can be computed using Equation (3). Equation (4) can be used to calculate the equivalent pressure zone height *x*.
(3)$$M=f_{c}b_{f}^{\prime }x\left(h-\frac{1}{2}x\right)+f_{s}^{\prime }A_{s}^{\prime }\left(h-a_{s}^{\prime }\right)+\sigma_{p}^{\prime }A_{p}^{\prime }\left(h-a_{p}^{\prime }\right)-\frac{1}{2}f_{t,p}\left[b{\left(h-h_{1}\right)}^{2}-h_{f}^{2}\left(b_{f}-b\right)\right]-f_{s}A_{s}a_{s}-\sigma_{p}A_{p}a_{p}$$
(4)$$f_{c}b_{f}^{\prime }x+f_{s}^{\prime }A_{s}^{\prime }+\sigma_{p}^{\prime }A_{p}^{\prime }=f_{t,p}b\left(h-h_{1}-h_{f}\right)+f_{t,p}b_{f}h_{f}+f_{s}A_{s}+\sigma_{p}A_{p}\;$$
where *M* is the calculated bending moment; *f*_c_ is the compressive ultimate strength of NC; *x* is the height of the equivalent compression zone; *b* is the thickness of the web; *h* is the height of the section; *h*_1_ is the height of the NC layer of the composite beam; *b*′_f_, *b*_f_ are the widths of the upper and lower flange plates of the section; *h*′_f_, *h*_f_ are the thicknesses of the upper and lower flange plates of the section; *f*′_s_, *f*_s_ are the calculated strengths of the longitudinal reinforcement in the compression and tension zones, respectively; *A*′_s_, *A*_s_ are the areas of the longitudinal reinforcement in the compression and tension zones, respectively; *a*′_s_, *a*_s_ are the heights of the longitudinal reinforcement in the compression and tension zones from their respective edges; *σ*′_p_, *σ*_p_ are the stresses of the prestressing tendons in the tensile and compressive zones when the beam is damaged and are equal to the tensile strength; *A*′_p_, *A*_p_ are the areas of prestressing tendons in the compressive and tensile zones, respectively; and *a*′_p_, *a*_p_ are the heights of the prestressing tendons from their respective edges in the compression and tension zones.

In a similar way to the III-I section, the normal section strength and equivalent section pressure zone height of the remaining five ‘I’ type sections can be calculated based on the internal force balance of each section.
(5)$$M=M_{c}+M_{s}+M_{P}+M_{u}$$
(6)$$\int_{A_{c}}f_{c}dA+f_{s}^{\prime }A_{s}^{\prime }+\sigma_{p}^{\prime }A_{p}^{\prime }=\int_{A_{u}}f_{t,p}dA+f_{s}A_{s}+\sigma_{p}A_{p}$$
(7)$$M_{c}=\int_{h_{c}}\int_{A_{c}}f_{c}dAdy$$
(8)$$M_{s}=f_{s}^{\prime }A_{s}^{\prime }\cdot \left(x_{0}-a_{s}^{\prime }\right)-f_{s}A_{s}\cdot \left(h-x_{0}-a_{s}\right)$$
(9)$$M_{p}=\sigma_{p}^{\prime }A_{p}^{\prime }\cdot \left(x_{0}-a_{p}^{\prime }\right)-\sigma_{p}A_{p}\cdot \left(h-x_{0}-a_{p}\right)$$
(10)$$M_{u}=\int_{h_{p}}\int_{A_{u}}f_{t,p}y_{p}dAdy$$
where *M*_c_ is the moment of the edge of the tensile zone of NC material; *M*_s_ is the moment of the edge of the tensile zone of the vertical stress reinforcement; *M*_p_ is the moment of the edge of the tensile zone of the prestressing tendons; *M*_u_ is the moment of the edge of the tensile zone of UHPC; *A*_c_ is the area of the NC compressive zone; *h*_c_ is the distance between the NC compressive zone and the bottom of the beam; *A*_p_ is the area of UHPC tensile zone; and *h*_p_ is the distance between the UHPC tensile zone and the bottom of the beam.

### 2.3. Cracking Moments of NC-UHPC Composite Beams

When a bridge structure cracks, the tensile zone is usually already in the elasto-plastic stage, which is too complicated to calculate directly for engineering purposes. For this reason, a plasticity correction factor *γ* is introduced to correct the elastic-plastic analysis when calculating the cracking moment of a prestressed member. The product of the ultimate tensile stress at the edge of the tensile zone and the elastic resistance moment of the transformed section is used as the calculated value of the cracking moment [41]. Therefore, the cracking moment of an NC-UHPC composite beam can be calculated with Equation (11).
(11)$$M_{\mathrm{cr}}=f_{t}W_{0}$$
where *M*_cr_ is the cracking moment; *f*_t_ is the ultimate tensile stress of concrete at the edge of the tensile zone of UHPC; and *W*_0_ is the resisting moment of inertia:(12)$$W_{0}=I_{0}/y_{0}$$
where *I*_0_ is the section’s moment of inertia, and *y*_0_ is the distance from the neutral axis to the bottom of the section.

The tensile properties at the edge of the tensile zone of prestressed members are generally composed of two parts: the compressive stress caused by prestressing and the tensile strength of the concrete material [15]. The ultimate tensile strength of the UHPC structure at the edge of the tensile zone is calculated according to Equation (13).
(13)$$f_{t}=\gamma f_{t,p}+\left[N_{y}/A+\left(N_{y}\cdot e\right)/W_{0}\right]$$
where *N_y_* is effective prestressing; *e* is the eccentricity of the steel strands; *A* is the area of the section; and *γ* is the plasticity correction factor for the moment of resistance of the tensile edge. This can be calculated from the transformed moment of resistance *W*_0_ at the edge of the tensile stress and the area moment *S*_0_ of the area below the gravity axis of the transformed section against the center of the gravity axis.
(14)$$\gamma =2S_{0}/W_{0}$$

## 3. Experimental Methods

### 3.1. Design of NC-UHPC Composite Beam

The T girder and box girder are the most common beam section forms in railway engineering [42]. Moreover, in the calculation of flexural capacity, the box girder can also be decomposed into a T girder [43]. For this reason, the T girder is used in this text. According to Figure 2, in comparison to the three types of I and II sections, the NC layer replacement height of the III-I section is greater, which can reduce structure costs and improve engineering economy, under the condition of satisfying the section’s stress, and is easily applied in actual projects. Compared to the III-II and IV sections, III-I sections have a higher central axis height (inside the wing) and stiffness, which not only improves the section’s crack resistance, but also fully utilizes the excellent tensile properties of UHPC materials under the same load. Therefore, using the 24 m simply supported prestressed concrete T-beam commonly found in railways as a prototype, three different combinations of section forms of the prestressed NC-UHPC T-beam were designed and fabricated with similar relationships. All beams were of III-I section, and post-tensioned with a 1860-grade 1 × 7 Φs 15.2 mm steel strand arranged longitudinally at the bottom of the beam, with seven wires and a 15.2mm diameter and a tension strength of 1860 MPa. The lower part of the beam was made of UHPC, while the upper part was made of NC. Each beam measured 2200 mm in total length, 210 mm in beam height, 80 mm in web width, and 200 mm in flange width. Figure 4 depicts the section dimensions and arrangements of the prestressing steel bundles, where the NC layers in beams B1, B2, and B3 correspond to heights of 40 mm, 60 mm, and 80 mm, respectively, and each height has two beams.

The cement material used in the test for UHPC and NC was ordinary silicate cement of standard 52.5, the fine sand particle size was less than 2 mm, the UHPC coarse sand particle size was 25 mm, and the NC coarse aggregate particle size was 220 mm. The compressive strength of UHPC and NC were tested with three 100 × 100 × 100 mm specimens. The tensile strength of UHPC was assessed using a direct tensile test with three dog-bone specimens with a cross-section of 50 × 50 mm. The stress-strain curve of UHPC and NC are depicted in Figure 5 and the strength of the UHPC and NC is shown in Table 1. The beam-casting process can be divided into the following three steps: (i) making a T-girder mold based on the section dimensions of the beam and using the T-shaped skeleton to strictly fix the position of the steel strands; (ii) casting the bottom UHPC layer of the beam based on the ratio of UHPC materials and maintaining the beam with geotextile coverage; (iii) after 24 h of curing, demolding and chiseling of the UHPC surface, followed by pouring of the upper NC layer. Then, following 28 days of curing with a geotextile cover, the prestressing tensioning and bending damage tests were performed.

### 3.2. Test Content and Measured Points Layout

To obtain the strain and deflection during the prestressing tension and static load bending tests of the structure, the following measuring methods were used: (i) pressure measurement—pressure transducers were arranged under the prestressing anchor and the hydraulic jack, respectively, to test the magnitude of the load during the tensioning and bending damage process; (ii) deflection measurement—three dial indicators were arranged in the 1/4 and mid-span sections of the beam to obtain the deflection values in different states of the 1/4 and mid-span sections during tension and bending damage; (iii) strain measure—eight strain gauges were arranged at one side of the mid-span along the height; the concrete strain was measured by a static strain gauge to obtain the distribution pattern of the longitudinal strain along the height during the tensioning and bending damage process. Figure 4 depicts the location of the beam strain gauges and the dial indicator arrangement.

### 3.3. Tensioning

To facilitate anchorage and uniform stress of the pressure sensor before tensioning, wedge-shaped blocks were set on both sides of the beam according to the inclination angle of each steel strand. At the same time, each beam was subject to the single-end tensioning method to reduce retraction loss during anchoring; the control tension stress of each beam was 1300 MPa. Figure 6 depicts the tensioning process. Table 2 shows the tensioning results for each beam, where the loss value refers to the loss caused by anchor retraction during tensioning.

Table 2 shows that, among the six beams, the prestress loss of B3-2 was the smallest, at 61.43 MPa, and the prestress loss of B1-2 and B3-1 was the largest, at 145.43 MPa; the reversed deflection of B2-2 was the smallest, at 0.95mm, and the reverse arch value of B1-1 was the largest, at 1.13 mm. Furthermore, it can be seen from Table 2 that, with the exception of the incorrect strain on the upper edge of B3-2, the tensile strain of the upper edge of B1-2 was the greatest, at 52.5 × 10^−6^, with a corresponding concrete tensile stress of 0.23 MPa; the compressive strain on the lower edge of B3-2 was −854.13 × 10^−6^, with a corresponding concrete compressive stress of −3.76 MPa, both of which meet the UHPC ultimate strength requirements of 8.5 MPa (tensile) and −131.9 MPa (compression), indicating that the beam prestress tension was effective and reasonable and can be used for static bending tests of prestressed NC-UHPC composite beams.

### 3.4. Static Bending Test

A 50 t load reaction frame combined with a steel I-beam was used to load the beam at two points in this test. The length of the beam’s pure bending section was 0.6 m, the bearing was 0.1 m from the beam end, and the effective span was 2 m. Figure 7 depicts the measuring point arrangement. The test was force-controlled. The force applied on the beam was measured by a pressure transducer under the hydraulic jack, with a load step of 10kN, loading until the beam broke.

## 4. Results and Discussion

### 4.1. Load-Deflection Curves and the Failure Process

Based on the results of the static bending test, the load-deflection curves of mid-span and 1/4 span sections of each beam were obtained; the results are shown in Figure 8. The load-deflection curves can be divided into four stages: the elastic stage (stage I), in which the load level is low and the deflection varies linearly with load increases; the uniform cracking stage (stage II), in which the variation rate of the deflection increases slightly with the load; the crack development stage (stage III), in which a gradual decrease in the load-deflection curve’s slope and an accelerated increase in the deflection occurs; and the yielding stage (stage IV), in which the load-deflection curve tends to be horizontal, while the load no longer grows and the deflection increases sharply. The control loads of each beam at each stage are shown in Figure 9.

Figure 9 shows that, as the NC layer height increases, each stage’s control loads have a slight tendency to decrease. This is because the joint surface of each beam is closer to the section’s neutral axis, allowing the material properties to be fully realized. At the same time, ordinary concrete with a compressive strength of 80 MPa can completely meet the compressive requirements of NC-UHPC composite beams. If UHPC is used for the entire section, there will inevitably be waste.

### 4.2. The Development of Cracks

Figure 10 depicts the evolution of the crack distribution of each beam.

Based on Figure 10, and the loading process of the beam, it can be seen that: (i) at stage I, the load level was low, the internal force of the beam section was small, and the beam body was not yet cracked; (ii) at stage II, when the load increased to the cracking load, the first micro-crack appeared near the bottom of each beam section, and the load continued to increase. The micro-crack slowly extended upward, but the crack did not expand horizontally and more micro-cracks appeared near the first micro-crack; (iii) at stage III, when the load continued to increase to the crack development load, the first crack at the bottom of the beam gradually became the main crack and gradually developed and penetrated to the NC-UHPC joint surface. The crack width increased, a large number of new cracks appeared and the beam flexed and deformed. As the speed increased, the sound of the steel fiber being pulled out could be heard; (iv) at stage IV, the load increased slowly, the main crack extended to the flange plate and developed horizontally at the height of the joint surface. New cracks were no longer generated, the load barely increased and the deflection increased sharply, indicating that the steel strand had yielded and the beam had reached the ultimate bearing state.

It can be seen from Figure 9 that the yielding and ultimate load of beam B3-1 was lower than that of the others. It is worth noting that the beam B3-1 exhibited a different failure phenomenon than the other beams in stages III and IV. That is, before the vertical crack penetrated to the joint surface, within a distance from the left end of the beam, penetration cracks appeared on the NC-UHPC joint surface, the joint surface cracks extended beyond the mid-span position, and the horizontal crack’s farthest extension had intersected the vertical crack. However, the mid-span deflection barely changed, indicating that shear failure of the composite beam joint surface preceded bending failure. This caused the beam to undergo an early collapse compared with the others and to show no obvious yielding phase. During the failure stage, the appearance of numerous fine cracks in the interfacial zone and reduction in the interfacial zone stiffness may explain this, consistent with the conclusion of [18].

### 4.3. Strain of Concrete

Figure 11 depicts the vertical distribution of the longitudinal strains of concrete of each beam under different loads based on the test results.

According to Figure 11, the strain varied approximately linearly with height changes at lower load levels. The same applied to the concrete on the compressive side at higher load levels, while the strain on the tensile side varied abruptly with height changes. This was because cracks appeared in the concrete of the tension zone at higher load levels, causing deformation or damage to the strain gauges there, resulting in distortion of the strain gauge data which did not reflect the true strain of the beam. In general, the longitudinal strains in the mid-span sections of different beams were consistent with the flat section assumption, according to the basic assumptions presented in Section 2.2.

According to Figure 3, the distribution of strain on the section can be regarded as a straight line. The relationship between the ordinate *y* and strain *ε* of the section can be expressed by the Equation (15).
(15)ε=ky+A
where *k* is the slope of the line, *A* is the strain of the bottom of the section, and *k* and *A* both reflect the magnitude of the strain on the section.

The strain on the section in Figure 11 can be fitted using Equation (15) and the variation of *k* and *A* with load on different sections can be obtained, as shown in Figure 12. It is clear that, similar to the load-deflection curves (Figure 8), the variation in *k* and *A* with load also shows three stages: the curves vary linearly at the beginning, followed by a decrease in the slope, and finally tend to be horizontal.

### 4.4. Normal Section Strength and Cracking Moment

For actual engineering structures, the theoretical formula based on the specific state of the structure leads to a large gap between the theoretical calculation results and the actual values due to the inability to reflect the variability in the loads, materials, and so on. The engineering reliability principle must be applied to ensure the theoretical formula has a safety reserve to ensure the reliability of the structure in service. As a result, when calculating the normal section strength and the cracking moment, the safety factor must be considered; Equations (16) and (17) must be satisfied.
(16)$$M_{\mathrm{um}}\geq \gamma_{u}\cdot M_{\mathrm{uc}}$$
(17)$$M_{\mathrm{crm}}\geq \gamma_{\mathrm{cr}}\cdot M_{\mathrm{crc}}$$
where *M*_um_, *M*_uc_ are the measured value and calculated value of the ultimate state bending moment, where *M*_uc_ is calculated from Equation (3); *γ*_u_ is the strength safety factor; *M*_crm_, *M*_crc_ are the measured value and calculated value of the cracking moment, where *M*_crc_ can be calculated with equation (11); and *γ*_cr_ is the safety factor of the crack. Figure 13 shows a comparison of *M*_uc_, *M*_um_, *M*_crm_ and *M*_crc_.

As can be seen from Figure 13, the measured value and the design value both decrease with the height of the NC layer, indicating that the NC layer has a negative effect on the normal section strength. The resistance factor *γ*_u_ ranges from 1.7 to 1.9 and *γ*_cr_ ranges from 1.5 to 1.7, which demonstrates that the calculation formulas for the ultimate state bending moment and the cracking moment proposed in Section 2 are accurate and have a safety reserve. It should be noted that the strength of the beam depends on the dimensions of the specimens in the test. It has been shown that, as the size increases, the flexural strength of UHPC beams tends to decrease, but the size effect on the flexural strength is negligible due to the high ductility of UHPC [10,44]; therefore, the normal section strength and cracking moment discussed here are reasonable up to a point.

It can also be seen in Figure 13 that the measured value of the cracking moment for the B3-2 beam was greater than for the other beams, which is because the actual tensioning stress value of B3-2 was the largest. This can be explained according to Equations (3) and (13) which indicate that the ultimate state bending moment is dependent on the strength of the steel strands, while the cracking moment is dependent on the tensioning stress.

## 5. Conclusions

The calculation formulas for the strength and cracking moment of NC-UHPC composite beam were deduced based on the principles of composite material mechanics and the railway bridge design code and verified by use of static bending tests with three different section beams. The following conclusions can be drawn from the research:

(1) The sections of NC-UHPC composite beams were classified into six types based on the relative position relationship of the composite beam’s compression zone, the thickness of the NC layer, and the thickness of the upper flange plate. Based on this, the calculation formulas for the cracking moment and the normal section strength of NC-UHPC composite beam were established in accordance with the current railway bridge design code.

(2) The test results for flexural performance of six NC-UHPC composite beams showed that flexural damage to the NC-UHPC composite beams exhibited four different stages: an elastic stage, a uniform cracking stage, a crack development stage and a yielding stage.

(3) The results for the longitudinal strain distribution in the mid-span of the six NC-UHPC composite beams showed that the beams conformed to the plane-section assumption during bending damage and the variety of strain resembled the load-deflection curves. Using NC with a compressive strength of 80 MPa instead of UHPC in the compression zone worked well and can be considered in engineering applications.

(4) The test values for each beam’s normal section strength and cracking moment showed that the strength of the steel strands affected the ultimate state bending moment, whereas the tensioning stress had an impact on the cracking moment. The formulas established in this paper can be used for the calculation of the normal section strength and the cracking moment of NC-UHPC composite beam with high accuracy and with a safety reserve.

It should be noted that the NC used in this study had a compressive strength of 80 MPa. In future research, the feasibility of using NC with a lower compressive strength instead of UHPC should be considered.

## Figures and Tables

**Figure 1 materials-16-00879-f001:**
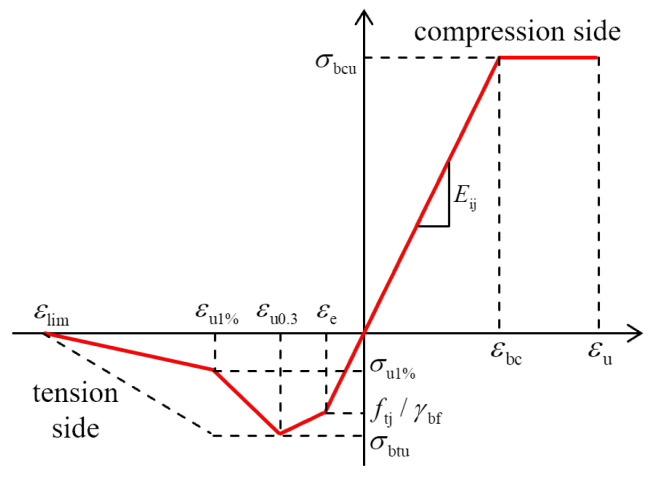
Stress–strain diagram for UHPC.

**Figure 2 materials-16-00879-f002:**
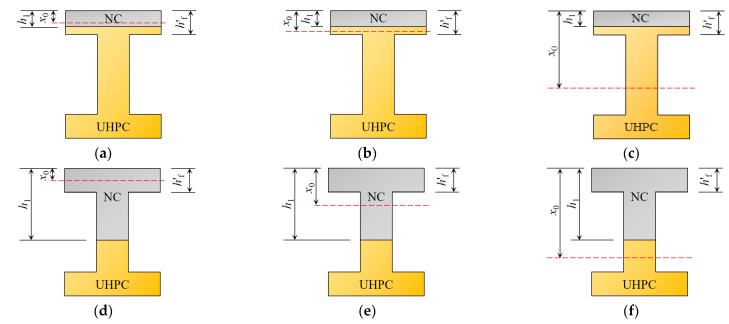
Type of section combination; (**a**) I; (**b**) II-I; (**c**) II-II; (**d**) III-I; (**e**) III-II; (**f**) IV.

**Figure 3 materials-16-00879-f003:**
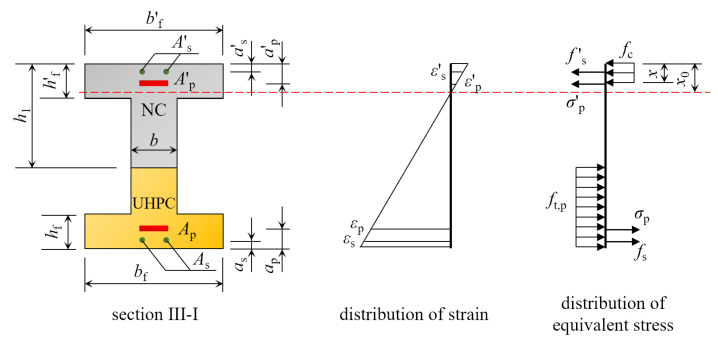
Calculation diagram for cross-section III-I.

**Figure 4 materials-16-00879-f004:**
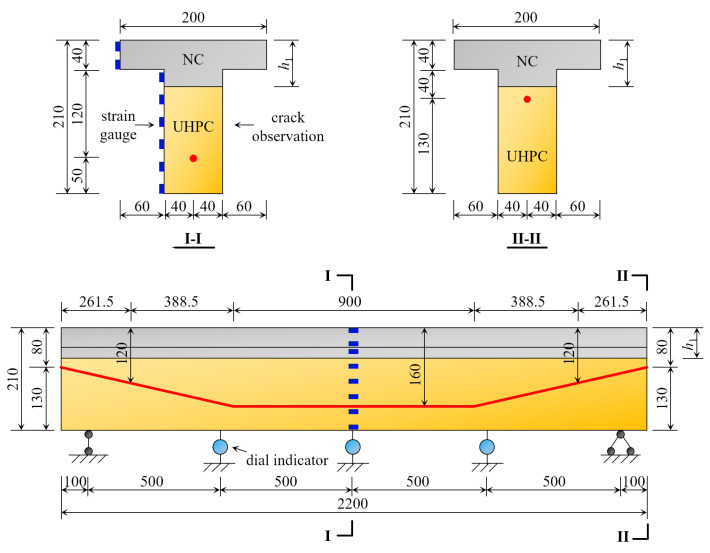
The size of the beam section and bundle arrangement (mm).

**Figure 5 materials-16-00879-f005:**
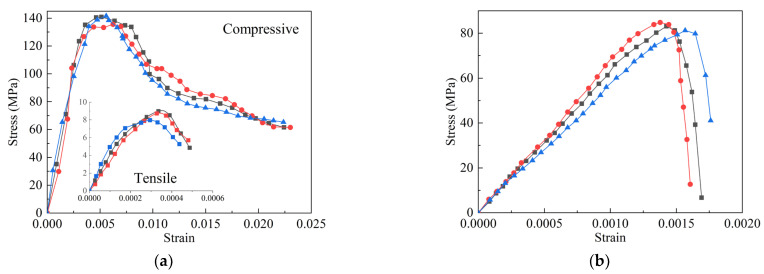
The stress-strain curve; (**a**) UHPC; (**b**) NC.

**Figure 6 materials-16-00879-f006:**
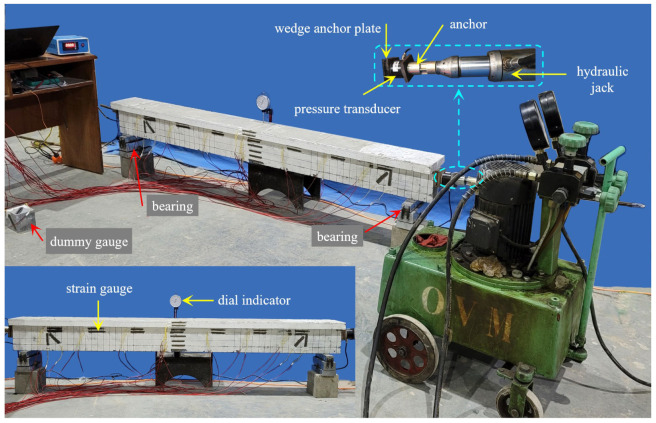
Schematic diagram of prestress tensioning.

**Figure 7 materials-16-00879-f007:**
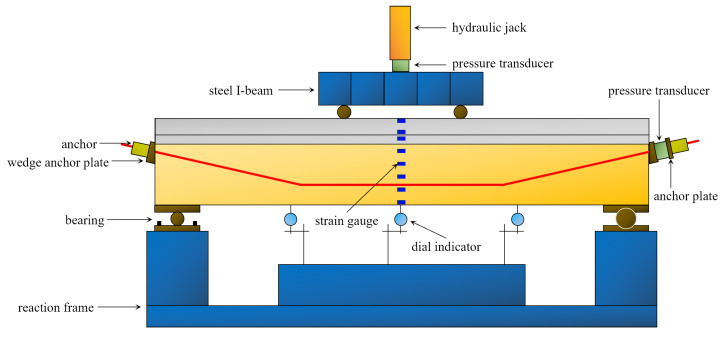
Loading diagram of bending test.

**Figure 8 materials-16-00879-f008:**
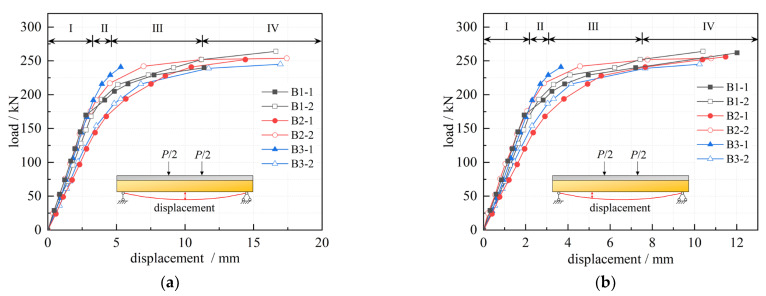
Load-deflection curves; (**a**) mid-span; (**b**) 1/4 span.

**Figure 9 materials-16-00879-f009:**
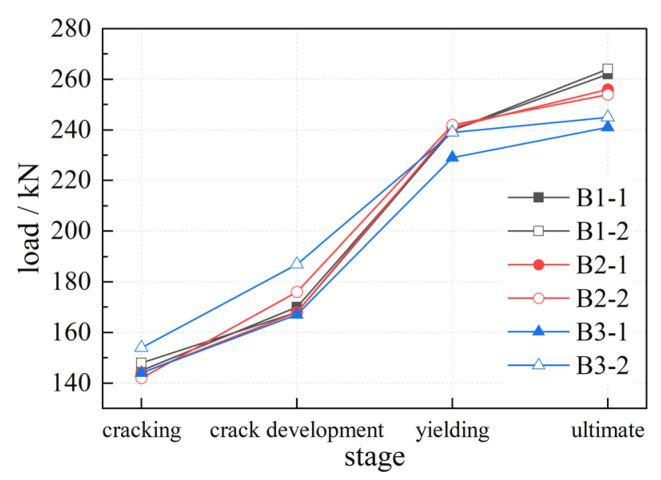
Control load of each stage of beams.

**Figure 10 materials-16-00879-f010:**
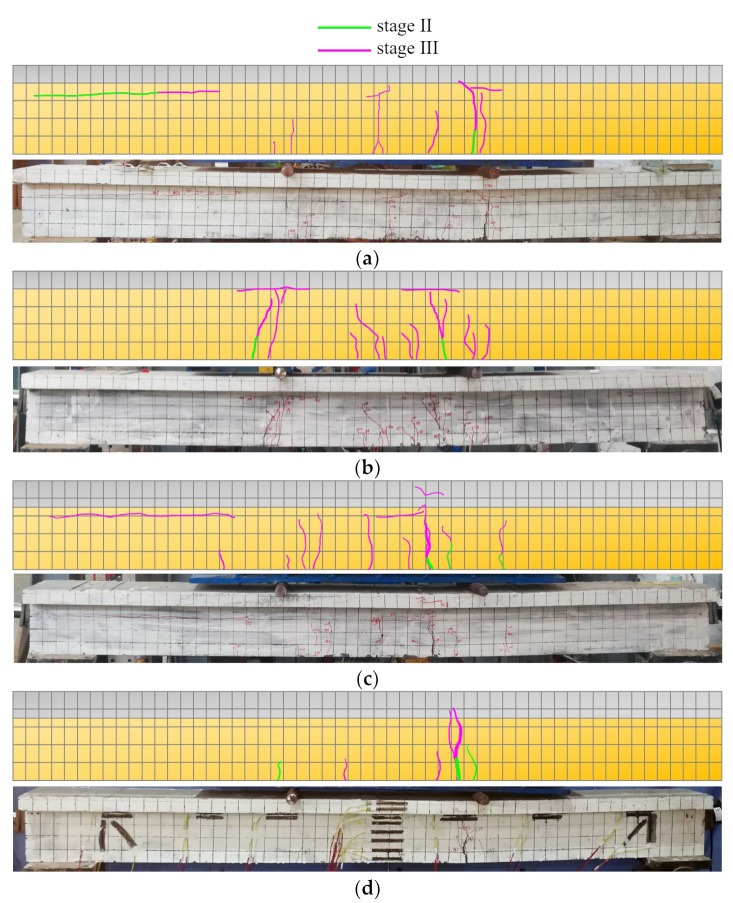
Crack distribution results for each beam; (**a**) B1-1; (**b**) B1-2; (**c**) B2-1; (**d**) B2-2; (**e**) B3-1; (**f**) B3-2.

**Figure 11 materials-16-00879-f011:**
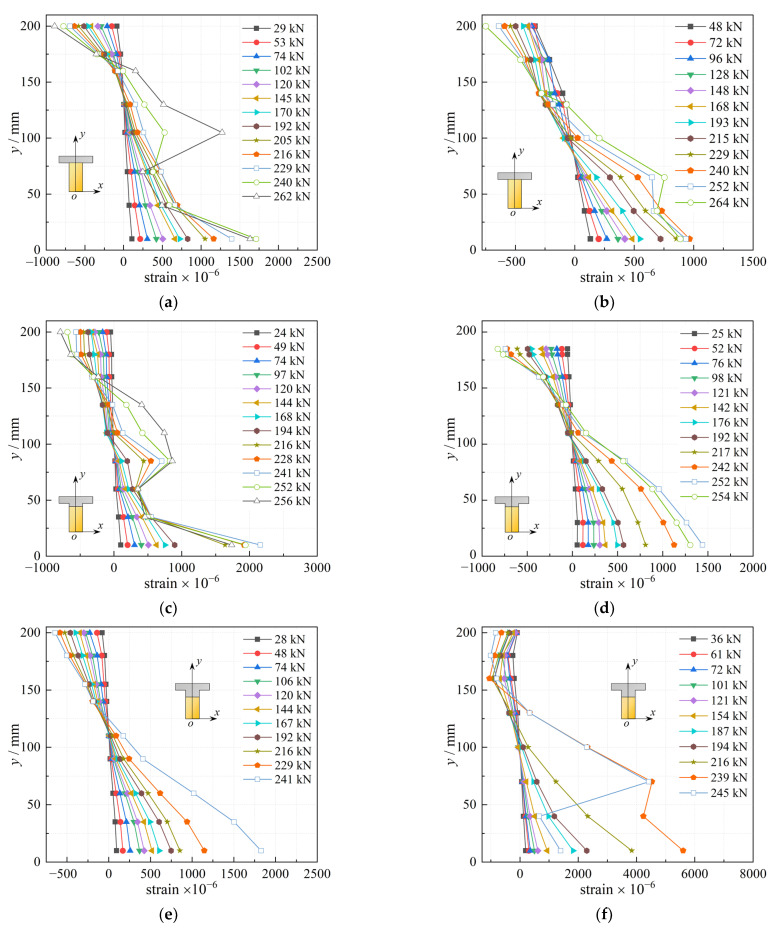
The result of the vertical distribution of the longitudinal strain; (**a**) B1-1; (**b**) B1-2; (**c**) B2-1; (**d**) B2-2; (**e**) B3-1; (**f**) B3-2.

**Figure 12 materials-16-00879-f012:**
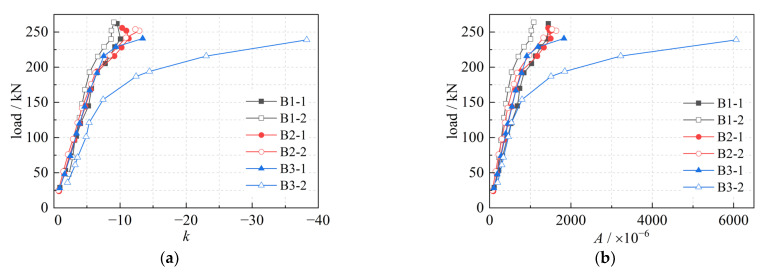
The variation in *k* and *A*; (**a**) *k*; (**b**) *A*.

**Figure 13 materials-16-00879-f013:**
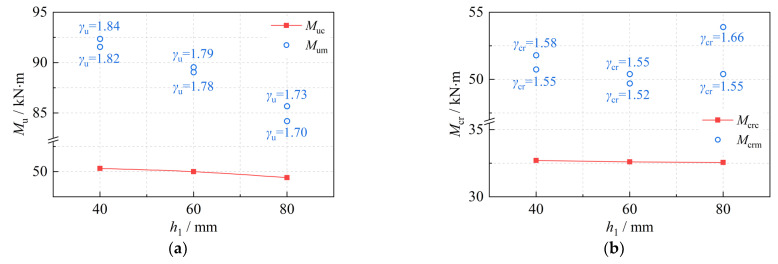
The calculated value and measured value of the normal section strength. (**a**) Ultimate state bending moment; (**b**) cracking moment.

**Table 1 materials-16-00879-t001:** Strength of the tested specimens.

Value	UHPC		NC
	Compressive	Tensile	
Tested value/MPa	140.79	8.97	83.15
	135.56	8.72	84.88
	141.55	7.97	81.26
Mean value/MPa	139.30	8.55	83.10
Coefficient of variation	2.34%	6.09%	2.18%

**Table 2 materials-16-00879-t002:** The results of the prestress tensioning of each beam.

Number of Beams	Tensioning Stress/MPa			Strain/×10^−6^		Reversed Deflection/mm
	Control Value	Actual Value	Loss Value	Upper Edge	Lower Edge	
B1-1	1300	1158.57	141.43	45.06	−498.07	1.13
B1-2	1300	1153.57	146.43	52.50	−458.75	0.97
B2-1	1300	1159.29	140.71	48.94	−490.91	1.08
B2-2	1300	1180.71	119.29	−1.01	−479.19	0.95
B3-1	1300	1153.57	146.43	50.00	−518.18	1.10
B3-2	1300	1238.57	61.43	−849.96	−854.13	1.02

## Data Availability

The data used to support the findings of this study are available from the corresponding author upon request.

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
