# Peer review of "Theoretical and Experimental Investigation on the Flexural Behaviour of Prestressed NC-UHPC Composite Beams"

_materials, 2023, doi:10.3390/ma16020879_

Round 1
Reviewer 1 Report
Review for materials-2119238-peer-review-v1
Title: Theoretical and Experimental Investigation on the Flexural Behaviour of Prestressed NC-UHPC Composite Beams
This study is based on the investigation of study the normal section strength and cracking bending moment of NC-UHPC composite beams. In addition, the calculation formulas of the cracking moment and bending capacity of NC-UHPC composite beams are derived by reference to the existing design specification of railway bridges and culverts. The experimental investigation was performed on six beams through a static bending test to explore the bending strength of the NC-UHPC composite beams.
The manuscript gathers theoretical and experimental results very interesting from a viewpoint of engineering design application and it presents a valuable contribution, possibly stimulating further research. This work is overall well-written, and the graphical representations are well-done.
Therefore, I have no hesitation in recommending it for publication in “Materials”. However, a Major revision is required, and some suggestions are proposed. To this end, the Authors are encouraged to prepare a revised version in which the following suggestions should be considered while finalizing their paper:
1) In the introduction section, the authors should highlight the essentials of the state-of-art of numerical modeling strategies for simulating the fracture behavior in UHPC/UHPRC/UHPRFC since it could be also interesting the focus on the new nanotechnologies adopted in this field giving better performance also from the bonding strength viewpoint between concrete/steel or between different concrete mixtures. Thus, for a better understanding of the above-mentioned aspect, the authors are encouraged to discuss in this Section the existing numerical methods able to predict and detect the damage phenomena in UHPC and the bibliographic context of the paper could be properly enlarged by discussing these works and, if necessary, other works cited therein:
a) Failure analysis of ultra high-performance fiber-reinforced concrete structures enhanced with nanomaterials by using a diffuse cohesive interface approach, Nanomaterials 10 (9), 1792, http://dx.doi.org/10.3390/nano10091792
b) Experimental and numerical study on mechanical properties of Ultra High Performance Concrete (UHPC), Construction and Building Materials, Volume 156, 15 December 2017, Pages 402-411. https://doi.org/10.1016/j.conbuildmat.2017.08.170
2) There are some words in the text with the symbol “-” breaking the words. (i.e. contri-bution at line 93, ten-sile at line 97, en-gi- at line 185 etc.). Please carefully check the whole Manuscript.
3) Fig. 2c seems it be the same as Fig.2b, probably the authors forgot to move the dotted red curve over the upper flange.
4) The Figure 2 caption seems to be wrong, check the nomenclature of the section combinations.
5) I got confused by the nomenclature of the beams at lines 187-191. Maybe due to the typos in the captions of Fig.2, so I suggest to the authors fix these issues and, if necessary, consider adopting a more clear nomenclature.
6) Section 3.2 is heavily convoluted and repetitive. “three tests were performed during the tests” or 1/4 and midspan are reported repetitively 3 times.
7) Due to the lack of clarity on the nomenclature of the beams, section etc. it is not clear the difference between the B1-1 and B1-2. Are they 2 beams with the same geometrical parameter?
8) In Fig.8 the control load of each stage is not in line with what was reported in Fig. 9. The beam B3-1 seems to be subjected to an early collapse compare with the others and doesn’t seem to be a yielding phase.
9) The photos reported in Fig. 9 were not taken rigorously. In some of these, the beam is cut and the angle is different for each photo. For the next experimental investigations, I suggest setting a camera in a fixed position frontal to the beam.
10) Regarding what was reported in Line 297, how the readers could see what happened in terms of crack evolution from Figure 9 if there are not reported the photos at the different stages? They seem to be the photos at the final stage.
Author Response
Dear reviewer,
Thank you very much for your valuable comments on this manuscript, your comments provided great help for us to improve this manuscript. The authors have revised the manuscript according to your comments, please see the attachment for more details.
We would like to thank the referee again for taking the time to review our manuscript.

Reviewer 2 Report
The manuscript entitled "Theoretical and Experimental Investigation on the Flexural Behaviour of Prestressed NC-UHPC Composite Beams” presented theoretical and experimental analyses for NC and UHPC prestressed composite beams.
The manuscript lacks clarity and cannot be accepted in the current format. The main purpose of this manuscript is unknown and this should be highlighted. I highly suggest a major review and submission for re-review.
Comments:
1- Appreciations should be defined at the first call and then they can be used throughout the text.
2- The abstract is a little short. The methodology used in this experimental work should be highlighted. What are the investigated parameters?
3- Lines 48-51: One of the advantages of using UHPC is the high compressive strength. Concrete is excellent in compression and very weak in tension. It is common to use concrete to support the compressive stresses and steel reinforcement for tensile stresses. So, why did the authors mention that UHPC in the compression zone can be replaced with NC? This point should be highlighted in more detail and explanations.
4- Line 72: The authors confirm that the tensile strength of UHPC is seldom to be considered. That means the contribution of concrete in the tension zone is neglected. So, why did the authors replace UHPC from the compression zone with NC? The novelty of this manuscript should be better addressed in the introduction section.
5- Line 92: The authors did not mention anything about using steel fibers in the abstract or even in the introduction.
6- Lines 113-114: Why the concrete in the compression zone cannot be fully used?! This point should be highlighted by the authors.
7- Lines 118-119: The authors in lines 63-71 mentioned more details about the interfacial zone between the NC and UHPC and how it will control the behavior at the ultimate state. However, this assumption will affect the theoretical results at the ultimate state. Moreover, the authors did not provide detailed verifications for the theoretical results.
8- The compatibility between the prestressed tendons and the surrounded concrete should be provided in the assumptions.
9- Lines 139-140: This reviewer thinks that iterations should be used to solve equation 4. Is that correct and how did the authors solve this equation to find the stress block depth (x)?
10- Lines 149-150: May I ask why the authors mentioned using prestressed tendons in the compression zone? Is it normal to use prestressing forces in compression? Also, did the authors consider the incremental increase in the prestressing force during loading? Which prestressed tendon profile did the authors develop in this process? Was it a straight or trapezoidal profile?
11- Line 172: What do you mean by "moment of resistance"? How did the authors calculate this term?
12- Section 3.1: The experimental stress-strain relationships of concrete (NC and UHPC) in tension and compression should be provided. Moreover, more details about the material properties of the used prestressed tendons.
13- Lines 238-239: Did the authors consider the losses in the developed theoretical process?
14- Section 3.4: Were the static tests forced-controlled or displacement-controlled?
15- Section 4.4: The authors provided well-known equations to calculate the design strength and cracked moment on this kind of beam. However, the authors did not provide detailed verification of the provided theoretical results. I highly recommend verifying the load-deflection relationships as well as the strain distributions.
Author Response

(The authors gave the same response as above.)

Author Response
Dear Reviewer:
Thank you very much for taking the time to review our manuscript and for your recognition of our manuscript.
Round 2
Reviewer 1 Report
The Authors have carefully considered the comments and the suggested changes.
They did their best to address every one of them.
I am strongly satisfied with their work and I think that in this form the Manuscript meets the standards required by the journal.
Author Response
Dear Reviewer:
Thank you so much for taking the time to review our manuscript and for your recognition of our manuscript.
Reviewer 2 Report
This manuscript still lacks clarity, especially after the authors' response. It can not be accepted in the current format.
1- The authors MUST respond to the comments and mention where to find the answer in the revised manuscript. Do not let the reviewer guess where to find your response in the revised manuscript.
2- Points 4 and 6: "The authors confirm that the tensile strength of UHPC is seldom to be considered. That means the contribution of concrete in the tension zone is neglected. So, why did the authors replace UHPC from the compression zone with NC?" and "Why the concrete in the compression zone cannot be fully used?! This point should be highlighted by the authors". The response to these comments is not clear. This reviewer can say "If the stress in the compressed zone is much less than the compressive strength of UHPC, which cannot be fully used, the authors can use much steel reinforcements or prestressed tendons in the tension zone to increase the beam capacity and economically get all benefits from the cross-section in the compression zone. Which is more economic, using UHPC in the tension zone to use the tensile strength or using steel reinforcement or pre-stressing in the tension zone?
3- Point 7: The authors provided very simple theoretical calculations by ignoring the relative slip between the NC and UHPC and the bond and slip between UHPC and steel strand.
4- Point 8: The authors mentioned: "the bond and slip between UHPC and steel strand are ignored". How this? The authors ignored the bond and the slip at the same time. I can understand that you can assume a full bond. So, you ignored the slip. How did you ignore the bond and slip between UHPC and the steel strand? That means there was no contribution from the steel strand to the cross-section.
5- The authors mentioned in the abstract "To study the normal section strength and cracking bending moment of NC-UHPC composite beam, based on the current railway bridge design code, calculation formulas are established considering the tensile strength of UHPC" and "The calculation formulas of the normal section strength and cracking bending moment of NC-UHPC composite beam were verified by the test". This means that the submitted manuscript developed and verified these formulas. However, the authors responded to point 15: "The main purpose of this paper is making the comparison of the test results and theoretical results, so not provided validation of the equations here". Is that logic?
Author Response
Dear reviewer,
Thank you so much for taking the time to review and point out the problem in our manuscript. The authors have revised the manuscript according to your comments, please see the attachment for more details.
